# In-Depth Insight into the Mechanism of Incorporation of *Abelmoschus manihot* Gum on the Enhancement of Gel Properties and In Vitro Digestibility of Frankfurters

**DOI:** 10.3390/foods12071507

**Published:** 2023-04-03

**Authors:** Dongxue Yuan, Xue Liang, Baohua Kong, Fangda Sun, Xin Li, Chuanai Cao, Qian Liu

**Affiliations:** 1College of Food Science, Northeast Agricultural University, Harbin 150030, China; 2Sharable Platform of Large-Scale Instruments & Equipments, Northeast Agricultural University, Harbin 150030, China; 3Heilongjiang Green Food Science & Research Institute, Harbin 150028, China

**Keywords:** frankfurters, *Abelmoschus manihot* gums, gel properties, conformational characteristics, in vitro digestibility

## Abstract

This study aimed to investigate the effects of different concentrations (0.1, 0.2, 0.3, 0.4, and 0.5% *w/w*) of *Abelmoschus manihot* gum (AMG) on the gel properties and in vitro digestibility of frankfurters. The results indicated that AMG incorporation significantly enhanced the emulsion stability and texture of frankfurters, as well as the dynamic rheological characteristics of raw meat batter, with the optimal concentration being 0.3% (*p* < 0.05). Furthermore, hydrogen bonds and disulphide bonds were the main molecular forces of the frankfurters in the presence of AMG. Microstructural images showed that more uniform and dense microstructures of frankfurters were formed due to AMG supplementation. In addition, AMG incorporation significantly increased the in vitro protein digestibility of frankfurters as the level of addition increased (*p* < 0.05). In conclusion, our results provided critical information for the practical application of AMG in the production of emulsified meat products.

## 1. Introduction

Frankfurters, one of the most popular emulsified meat products, are widely preferred and consumed worldwide due to their unique flavour and excellent nutritional value [1]. However, some crucial processing parameters (e.g., pH value of raw meat, lean meat to fat ratio, water content, ionic strength, heating method, and heating temperature) might lead to the quality deterioration (especially for relatively weaker water/fat holding capacities and looser textural properties) of the final commercial frankfurters to some extent, which subsequently affects the consumer’s purchasing inclination and preference. Thus, to effectively address the aforementioned issues in the actual processing of frankfurters, numerous functional additives (e.g., starch, hydrocolloids, cellulose, plant proteins, and animal proteins) and some emerging processing technologies (e.g., ultrasound treatment, high-pressure processing, electromagnetic field treatment, and pulsed electric field treatment) have been employed to increase the cooking yield and enhance the quality properties of the final meat products [2,3,4]. Among the above strategies, the incorporation of hydrocolloids to enhance the quality attributes of emulsified meat products is considered the most economical, convenient, and effective method in the meat industry. Jiang, Cao, Xia, Liu, and Kong (2019) demonstrated that the incorporation of curdlan gels could significantly facilitate the formation of a more uniform and dense gelation network structure of the myofibrillar protein (MP) after heat treatment, which subsequently significantly reduced the cooking loss and improved the textural characteristics of frankfurters [5]. Moreover, Cao et al. stated that the incorporation of κ-carrageenan evidently enhanced the cooking yield and emulsion stability, in addition to enhancing the texture of the frankfurters [6]. They also determined that the effects of κ-carrageenan application in frankfurters were greatly dependent on the addition levels and forms of κ-carrageenan, which was attributed to the different molecular interactions between proteins and κ-carrageenan. Li et al. also indicated that the incorporation of deacetylated konjac glucomannan (KGM) could effectively promote the transition of myosin from α-helix to β-sheet or β-turn, which subsequently led to a higher gel strength of heat-induced myosin gels [7]. Therefore, numerous hydrocolloids have been considered as potential and efficient functional non-meat additives to promote the quality attributes of meat products, gaining increasing interest in the meat industry.

*Abelmoschus manihot* gum (AMG), a new type of natural hydrocolloid, is mainly extracted from the root and stem of *Abelmoschus manihot* (a plant of the Malvaceae family), and is mainly composed of arabic polysaccharide and galactopolysaccharide. As a relatively rich renewable natural polymer resource, AMG is commonly used as a thickener and stabiliser in the modern food industry owing to its biocompatibility and safety characteristics. Moreover, Li et al. (2022) indicated that AMG exhibited a superior swelling ratio and viscosity, leading to its excellent water-holding capacity [8]. Based on this aspect, we hypothesised that AMG would show a greater potential to reduce cooking loss and thus enhance the emulsion stability of frankfurters. However, to the best of our knowledge, no information was available on the application of AMG in emulsified meat products. Moreover, previous studies reported that the incorporation of hydrocolloids had various effects on the in vitro digestibility of meat proteins. For instance, Cao et al. (2022) indicated that the incorporation of κ-carrageenan clearly decreased the in vitro digestibility of meat proteins in frankfurters in the gastric and intestinal tract [9]. However, Saengsuk et al. found that the addition of low acyl gellan effectively promoted the protein digestibility of restructured pork steak [10]. In addition, the interactions between meat proteins and hydrocolloids have been seen to lead to the distinction of the meat protein gel network and affect the susceptibility of proteins to protease, resulting in the higher or lower protein digestibility of meat products [11,12]. Thus, we also assumed that the incorporation of AMG, especially for the higher addition levels, could lead to some changes in the in vitro protein digestibility of frankfurters. However, to the best of our knowledge, no information on this aspect was available.

Therefore, this study aimed to investigate the effects of the addition of different levels of AMG on the gel properties, conformational characteristics, and microstructures of frankfurters, as well as the dynamic rheological behaviour of raw meat batters. Moreover, changes in the in vitro digestibility of frankfurter meat proteins induced by the addition of AMG were also detected under simulated gastrointestinal tract conditions to further explore the nutritional values of AMG-containing frankfurters.

## 2. Materials and Methods

### 2.1. Materials and Chemicals

Post-rigor lean pork shoulder meat (moisture 73.90%, protein 21.36%, and fat 3.17%, based on total weight) and pork back fat (moisture 7.46% and fat 86.95%, based on total weight) from northeast min pig were purchased from Beidahuang Meat Co., Ltd. (Suihua, Heilongjiang, China), stored at 4 °C while delivered to the lab, and used within the same day. Food-grade AMG was kindly provided by Huaxing Food Additives Co., Ltd. (Jinan, Shandong, China). Pepsin (P7012, ≥400 units/mg) and trypsin (S31655, ≥2500 units/mg) were purchased from Sigma-Aldrich (St. Louis, MO, USA). All spices were purchased from Vemis Spices Co., Ltd. (Taizhou, Jiangsu, China).

### 2.2. Preparation of Frankfurters

The fundamental formulations of the different frankfurters are shown in Table 1. Different levels of AMG incorporation, i.e., 0.0, 0.1, 0.2, 1.3, 0.4, and 0.5%, *w/w*, based on the total weight of the samples, were added into meat batters and marked as Control, AMG-0.1%, AMG-0.2%, AMG-0.3%, AMG-0.4%, and AMG-0.5%, respectively. The detailed processing procedure of the frankfurters was performed according to the method of Yuan et al. [13]. Briefly, all visible connective tissue was removed from the raw meat before the preparation of the meat batters. Then, the pork shoulder meat and pork back fat were ground through 8 mm and 3 mm plates, respectively, via a mincer (BJRJ-82, Xinghe Mechanical Co., Ltd., Nantong, Jiangsu, China). The lean pork meat was then added to the bowl chopper (BZBJ-20, Xinghe Mechanical Co., Ltd., Nantong, Jiangsu, China) and chopped for 2 min with salt, sodium nitrite, composite phosphates, and half of the ice. Then, the pork back fat, different concentrations of AMG, spices, and the remaining ice were added to the meat batters and chopped for an additional 3 min. After that, sodium erythorbate was added to the meat batters and chopped for 1 min. The entire temperature of meat batters was below 12 °C in all instances. The meat batters were stuffed into collagen casings with 20 mm diameter. The frankfurters were then transferred to an automatic smoking chamber (BYXX-50L, Xinghe Mechanical Co., Ltd., Nantong, Jiangsu, China) for drying and smoking. All the different prepared frankfurter samples were vacuum packed and stored in the refrigerator at 4 °C until the instrumental measurements were finished.

### 2.3. Cooking Loss and Emulsion Stability

The cooking loss (%) of the different frankfurters was determined using the method of Cao et al. [9]. Briefly, the cooked frankfurters (length and diameter of 15 cm and 20 mm, respectively) were refrigerated overnight at 4 °C and then weighed. The cooking loss (%) was expressed by the weights of cooked frankfurters relative to the weights of un-cooked frankfurters, and the calculation formula was as follows:$$Cooking\;loss(\%)=\frac{W_{1}-W_{2}}{W_{1}}\times 100$$
where W_1_ was the weight of un-cooked frankfurters (g) and W_2_ was the weight of cooked frankfurters (g).

The emulsion stability of the different frankfurters was measured and expressed according to the same procedure described by Zhang et al. [14]. Briefly, approximately 35 g of meat batter was weighed in tubes and then centrifuged (3500× *g*, 5 min, at 4 °C). Subsequently, the tubes were heated at 80 °C in a water bath for 30 min and then left to stand upside down for 1 h to release juice into the pre-weighed aluminium dish. Then, the aluminium dish was weighed and placed in the 105 °C baking oven (DHG-9030A, Xinghe Mechanical Co., Ltd., Nantong, Jiangsu, China) and baked until the weight was constant. The expression of emulsion stability was as follows: the total released liquid (%) was expressed as the ratio of the weight of the released juice after inversion for 1 h to the total weight of the meat batter. Released water (%) was expressed as the ratio of weight lost after baking to the total weight of the meat batter. The released fat (%) was expressed as the ratio of the weight left after baking to the total weight of the meat batter.

### 2.4. Texture

According to the detailed testing procedures and parameters of Yuan et al. [15], two deformation tests were used to determine the texture of frankfurters with a TA-TX plusC texture analyser (Stable Micro Systems Co., Ltd., Godalming, UK). The detailed testing parameters were as follows: the pre-test speed, test speed, and post-test speed were 1.5 mm/s, 1.5 mm/s, and 10.0 mm/s, respectively, and the trigger force was 15.0 g. Moreover, the testing procedure was divided into two consecutive cycles as follows: (1) the first cycle attempted to break the surface of the frankfurters at 15.0% strain, which mainly reflected their hardness (g), springiness (%), and resilience (g) and (2) the second cycle punctured the interior of the frankfurters at 75.0% strain, which mainly reflected their chewiness (g.s), fracturability (g), and adhesiveness (g.s). The holding time between two cycles was 5 s.

### 2.5. Dynamic Rheological Measurements

The dynamic rheological properties of different uncooked meat batters during the heat treatment were measured according to the methods of Cao et al. using a rheometer (DHR-1, TA Instruments Inc., New Castle, DE, USA) with a 40 mm parallel plate and a 0.5 mm gap [9]. Before performing formal temperature ramp sweep tests, the linear viscoelastic region (LVR) variation range of the samples at different heating points (such as 20, 30, 40, 50, 60, 70, and 80 °C) was performed using amplitude sweep tests at a frequency of 1.0 Hz. The results indicated that a strain of 0.012 would ensure that all the samples could be measured within the LVR. Finally, formal temperature ramp sweep tests were conducted from 20 °C to 80 °C at a fixed heating rate (1 °C/min), maintained at 80 °C for 5 min, and then cooled from 80 °C to 20 °C at a fixed cooling rate (2 °C/min) using a circulating water bath at a fixed frequency (1.0 Hz) with a strain of 0.012. The storage modulus (*G*′), loss modulus (*G*′′), and loss tangent (tan *δ*, *G*′′/*G*′) were recorded to assess the rheological properties of the different meat batters.

### 2.6. Fourier Transform Infrared (FTIR) Spectroscopy

The FTIR spectra of the freeze-dried frankfurters were recorded using the method of Sun, Zhou, Zhao, Yang, and Cui via a Nicolet is50 FTIR spectrometer (Nicolet, Germany) with a scanning range of 4000–400 cm^−1^ [16]. Subsequently, the amide bands observed in the 1600–1700 cm^−1^ range were deconvoluted and curve-fitted using PeakFit v4.12 software (Systat Software Inc., San Jose, CA, USA) to analyse the secondary structure compositions (expressed as the α-helix, β-sheet, β-turn, and random coil content) of the meat proteins in the frankfurters [17].

### 2.7. Molecular Forces

The molecular forces of the frankfurters were determined and calculated according to the detailed method described by Cao et al. [9]. Four different reagents were dissolved in 50 mmol∙L^−1^ sodium phosphate buffer (pH 7.0) to measure the corresponding molecular forces as follows: (a) 8 mol∙L^−1^ urea to measure the number of hydrogen bonds, (b) 0.5% (*w/v*) sodium dodecyl sulfate (SDS) to measure the number of hydrophobic interactions, (c) 0.25% (*v/v*) β-mercaptoethanol (β-ME) to measure the number of disulphide bands, and (d) 0.5 mol∙L^−1^ sodium thiocyanate (NaSCN) to measure the number of electrostatic interactions.

### 2.8. Scanning Electron Microscopy (SEM)

The microstructures of the different frankfurters were observed by SEM (S-3400 N, Hitachi, Ltd., Tokyo, Japan). The detailed pre-treatment method of the samples was performed as described by Cao et al. [6].

### 2.9. In Vitro Gastrointestinal Digestion

The frankfurters were minced and mixed with phosphate-buffered saline (5 mL, 10 mmol/L Na_2_HPO_4_–NaH_2_PO_4_, pH 7.0). The mixed solution was treated with pepsin and trypsin to simulate the human gastrointestinal digestion process. Then, the steps of gastric and intestinal digestion were carried out according to the steps in Section 2.9.1 and Section 2.9.2, respectively. In addition, the specific formulations of the simulated gastric fluid (SGF) and simulated intestinal fluid (SIF) were prepared according to the method of Jiang et al. [18], and are listed Table 2.

#### 2.9.1. Gastric Digestion

The gastric digestion of frankfurters was carried out according to the procedure of Wang, Li, Zhou, Li, and Liu with slight modifications [19]. Before digestion, different ground frankfurters (5 g) were mixed with phosphate-buffered saline (5 mL, 10 mmol/L Na_2_HPO_4_–NaH_2_PO_4_, pH 7.0) and homogenised (22,000 r/min, 1 min, 4 °C) using a multifunctional food processor (JYL-C93T, Joyoung Co., Ltd., Jinan, China). Subsequently, pepsin solution (10 mL, dissolved in pre-formulated SGF) was dissolved at a final concentration of 6 mg/mL in the obtained sample solution (10 mL), and incubated at 37 °C for 2 h (200 rpm). The reaction was terminated by the pH inactivation of pepsin (adjusted to pH 7.0 with a 2 mol/L NaOH solution).

#### 2.9.2. Intestinal Digestion

The intestinal digestion of the samples was carried out according to the method of Wang et al. with slight modifications [19]. Briefly, a trypsin solution (20 mL, dissolved in pre-formulated SIF) was dissolved at a final concentration of 6 mg/mL in the chyme (20 mL) obtained in the simulated gastric digestion, and incubated at 37 °C for 2 h (200 rpm). The reaction was terminated by the thermal inactivation of trypsin (heating at 95 °C for 5 min).

### 2.10. Digestion Products Analysis

Digestion samples were collected after 30, 60, 90, and 120 min during the simulated gastric and intestinal digestion. To remove undigested proteins from the digestion mixture, an equal volume of trichloroacetic acid solution (15% *w/v*) was added and maintained at 4 °C for 12 h [20]. The blank control group was prepared with phosphate-buffered saline instead of frankfurters.

#### 2.10.1. Protein Digestibility

The in vitro digestibility of the meat proteins in frankfurters was measured according to the method of Cao et al. [9]. Briefly, the digested samples were centrifuged (10,000× *g*, 20 min, 4 °C) and then the supernatant was collected to measure the protein concentration by the Biuret method. The protein digestibility of the frankfurters was calculated using the following formula:$$\mathrm{Protein}\;\mathrm{digestibility}\;(\%)=\frac{W_{1}-W_{2}}{W_{0}}\times 100$$
where W_0_ is the protein content of the initial frankfurters, W_1_ is the protein content of the frankfurters after digestion, and W_2_ is the protein content of the blank group.

#### 2.10.2. Kinetic Modelling of In Vitro Protein Digestion

The kinetic modelling of in vitro protein digestion was carried out using ORIGIN software (2018, 64 Bit). The fractional conversion model is as follows:$$C=C_{f}(1-e^{-\mathrm{kt}})$$
where C (%) is the protein digestibility in the intestinal phase at any digestion time, C_f_ (%) is the estimated plateau at extended digestion times, k (min^−1^) is the reaction rate constant, and t (min) is the digestion time in the simulated intestinal phase.

#### 2.10.3. Particle Size of Digested Products

Based on the methods and parameters proposed by Cao et al. [9], a Nano ZS dynamic light scattering instrument (Malvern Instruments Ltd., Worcestershire, UK) was used to analyse the particle size of the chyme after the simulated gastrointestinal digestion of the frankfurters. Before determination, each sample was diluted to 1.0 mg/mL with distilled water.

#### 2.10.4. Super-Resolution Microscopy

Following the method and parameters of Feng et al. [20], microscopic images of the frankfurters before and after simulated gastrointestinal digestion were obtained with the DeltaVision OMX SR (General Electric Company, Healthcare, Boston, MA, USA). Before observation, Nile Blue solution (10.0 μL, 0.1% *w/v* in distilled water) was used to stain the proteins in the initial, gastric, and intestinal samples (0.5 mL). Samples were allowed to stand for 20 min to complete staining, and then 5 μL of the samples were placed on concave microscope slides with coverslips to prepare observation samples. The protein phase was observed using a conventional widefield imaging model with an excitation wavelength of 640 nm and a scanning range of 512 × 512 μm^2^.

### 2.11. Statistical Analysis

Three independent batches of frankfurters (replicates) were prepared on different days. For each batch, measurements of their related traits were performed in triplicate. All data were expressed as the mean ± standard deviations (SD). To assess the significance of the main effects (*p* < 0.05), a one-way analysis of variance (ANOVA) together with Duncan’s multiple comparisons was performed using SPSS Statistics 20.0 software (SPSS Inc., Chicago, IL, USA).

## 3. Results and Discussion

### 3.1. Gel Properties of Frankfurters

#### 3.1.1. Cooking Loss and Emulsion Stability

Cooking loss is one of the basic indicators of the quality attributes of meat products, reflecting the water and oil holding capacity of the frankfurters in this study. As shown in Table 3, the cooking loss of frankfurters with AMG added decreased significantly compared to the control group (*p* < 0.05), indicating that more water and oil were retained in the meat protein matrix due to the addition of AMG. With increasing AMG concentrations from 0.1% to 0.5%, the cooking loss of the frankfurters decreased by 13.90, 18.15, 49.81, 54.05, and 71.04%. The reason for this phenomenon is that AMG has superior water absorption and can absorb more water and fat, thus reducing cooking loss. In addition, the addition of AMG promoted the formation of a fine and uniform gel network structure, thus trapping more water and fat. Furthermore, Han et al. pointed out that inulin could alter the interaction between water molecules and MPs and strengthen the network structure of the protein gel, thus enhancing the water-holding capacity of the MP gel [21].

Emulsion stability is another important index to reflect the quality profile of frankfurters. As shown in Table 3, compared to the control group, the emulsion stability of frankfurters with AMG added was significantly enhanced, with lower total released liquid, released water, and released fat (*p* < 0.05), which was in agreement with the trend of cooking loss. Jiang et al. reported that curdlan gum significantly improved the emulsion stability of frankfurters, mainly because curdlan molecules fully swell to establish a new compact network structure during thermal processing, thereby retaining more water and fat molecules in a protein gel matrix [5]. Álvarez and Barbut also indicated that the incorporation of inulin improved the emulsion stability of cooked meat batters by reducing fat loss [22]. However, there was no significant difference in the total amount of released liquid and released water from the frankfurters when the AMG concentration was higher than 0.3% (*p* > 0.05), suggesting that 0.3% AMG could achieve a superior effect in improving the emulsion stability of frankfurters.

#### 3.1.2. Texture

Table 4 shows the textural parameters of the samples. Compared to the control group, the incorporation of AMG clearly increased the textural parameters of the frankfurters in terms of hardness, resilience, springiness, fracturability, chewiness, and adhesiveness (*p* < 0.05). This phenomenon might be because the incorporation of AMG increased the viscosity of the meat batters and altered the secondary structures of the MPs, which facilitated the formation of a solid gelation network, thereby improving the textural attributes of the frankfurters. Ayadi, Kechaou, Makni, and Attia determined that the addition of carrageenan clearly improved the hardness and cohesiveness of turkey meat sausages [23]. Schuh et al. also indicated that the firmness of emulsified sausages increased due to the incorporation of carboxymethyl cellulose and microcrystalline cellulose [3]. However, all textural parameters of the frankfurters showed a decreasing trend when the AMG concentration was higher than 0.3%, and even the resilience, chewiness, and adhesiveness values were lower than those of the control group when the AMG addition level reached 0.5% (*p* < 0.05). This could be explained by the fact that excessive levels of exogenous additives might produce an overfilling effect in the MPs matrix, which results in the loss of springiness and resilience in frankfurters [24]. Ghafouri-Oskuei et al. reported that higher concentrations of flaxseed powder led to decreased textural parameters (e.g., hardness, cohesiveness, and gumminess) of meat batter [25]. Yang, Keeton, Beilken, and Trout also indicated that konjac significantly reduced the firmness value of frankfurters at a higher concentration [26]. Therefore, the texture results implied that a suitable AMG addition standard is a feasible strategy to improve the textural features of frankfurters, especially at an AMG concentration of 0.3%.

#### 3.1.3. Dynamic Rheological Properties

The dynamic rheological behaviour of raw meat batters was evaluated by recording the storage modulus (*G*′), loss modulus (*G*′′), and loss tangent (tan *δ*), which reflect the structural variation of meat protein [27]. The *G′* value describes the elastic property of the gel samples, as shown in Figure 1A (heating process) and B (cooling process). In the heating process, the *G′* values of all samples started to increase at 40 °C, which was mainly caused by the unfolding of myosin heads [21]. Then, the *G′* values of the meat batters reached their maximum values at approximately 50 °C, where the temporary MP gel network structure was formed [9]. At a temperature range from 50 °C to 59 °C, the *G*′ values exhibited a decreasing trend due to myosin tail denaturation, indicating that the temporary MP gel network structure was wrecked [28]. Finally, the *G*′ values continuously increased until the temperature reached 80 °C, implying that an elastic gel network structure was formed owing to the complete denaturation and aggregation of MP [29]. Moreover, the *G*′ values of the meat batters with AMG added were clearly higher than those of the control group during the entire heating process, and the AMG-0.3% group showed the highest *G*′ value (*p* < 0.05) of all. Petcharat and Benjakul noted that the *G′* value of bigeye snapper surimi protein increased significantly due to the addition of gellan during thermal processing [30]. However, the *G′* values showed a downward trend when the amount of added AMG exceeded 0.3%, which might be attributed to MP depletion flocculation [31]. For the cooling process (Figure 1B), the *G′* values of all samples increased constantly with decreasing temperature, indicating that an irreversible MP gel had formed. Furthermore, the *G′* values of the meat batters containing AMG were significantly higher than those of the control group, which was similar to the heating process. This result is consistent with that of Chen et al., who noted that chicken breast myosin with κ-carrageenan showed a higher *G′* value than the control group during the cooling treatment [32].

The loss modulus (*G*′′) is often used to characterise the viscosity property of gelation samples. As displayed in Figure 1C,D, the *G′′* values had a similar variation trend as the *G′* values with increasing and decreasing temperatures. Moreover, the *G′′* values of raw meat batters with AMG added were significantly higher than those of the control group at 80 °C, and the AMG-0.3% group had the highest *G′′* value (*p* < 0.05) of all, indicating that AMG could effectively improve the viscosity behaviour of the meat proteins. In addition, Zhuang et al. reported that polysaccharides could lead to a higher apparent viscosity of MPs than the control group [33].

The loss tangent (tan *δ*) was obtained from *G′′*/*G′* and represents the viscoelastic properties of the gel samples. The elastic behaviour of the sample was more prominent when the tan *δ* value was closer to 0, while the viscous behaviour of the sample was more prominent when the tan *δ* value was closer to 1 [33]. The tan *δ* values of the samples are shown in Figure 1E (heating process) and Figure 1F (cooling process). In the heating process, the tan *δ* values decreased sharply with increasing temperature from 20 °C to 47 °C, which meant that the samples transited gradually from a viscous state to an elastic semi-solid substance [27]. At approximately 51 °C, a small peak in tan *δ* was observed due to myosin denaturation. Then, as the temperature was further increased to 80 °C, the tan *δ* curve showed a downward trend, possibly due to actin aggregation [6]. Moreover, the tan *δ* values of the samples incorporated with AMG showed a trend of initially decreasing and finally increasing, and the AMG-0.3% group had the lowest tan *δ* value at 80 °C, implying that AMG incorporation could improve the elasticity of frankfurters. For the cooling process, the tan *δ* values slightly increased with decreasing temperature, and the tan *δ* values of samples added with AMG were lower than that of the control group at 20 °C (except for the AMG-0.1% group), which was similar to the result of the heating process. In addition, MP denaturation and aggregation occurred during the heating stage, where the gel network was formed first, and the gel matrix was further strengthened during the cooling process [34].

### 3.2. Conformational Characteristics of Proteins

#### 3.2.1. Secondary Structures

FTIR spectroscopy is a useful tool to explore the secondary structure of protein side chains [35]. The FTIR spectroscopy image of the samples is shown in Figure 2. The absorption peak of the amide I bands from 1600 to 1700 cm^−1^ can be used to reflect the secondary structural conformation of the protein. Moreover, the secondary structure distribution of the amide I band is as follows: α-helix (1650–1660 cm^−1^), β-sheet (1618–1640 cm^−1^), β-turn (1660–1690 cm^−1^), and random coil (1642–1650 cm^−1^) [36]. As shown in Table 5, the α-helix content of the samples clearly declined with increasing AMG concentration, while the content of β-sheet, β-turn, and random coil clearly increased (*p* < 0.05). This phenomenon illustrates that AMG incorporation could alter the MP conformations of frankfurters. Xu et al. obtained similar results and indicated that deacetylated KGM could promote the conformational transformation of myosin from α-helix to β-turn and β-sheet, which was due to the realignment of intramolecular electrostatic forces in the KGM-myosin mixture system [37]. Additionally, Mi et al. indicated that the α-helix content of silver carp surimi increased with the incorporation of curdlan and κ-carrageenan, while the content of β-sheet and β-turn decreased [38]. Moreover, the α-helix content was negatively related to the hardness of frankfurters, while the β-sheet content was positively related to gel strength [39]. Zhang, Yang, Tang, Chen, and You also noted that the decrease in α-helix content and the increase in β-sheet content favoured the formation of a fine three-dimensional network structure of proteins [40], which was consistent with the texture result in this study.

#### 3.2.2. Molecular Forces

Force-disrupting agents were used to treat the samples and change the solubility of the protein, which could characterise the molecular forces that stabilise the protein structure [41]. As shown in Table 6, among the four different molecular forces, the content of hydrogen bonding was much higher than the other three, which indicates that hydrogen bonding plays a crucial role in stabilising the MP network structure. Diao et al. noted that hydrogen bonds predominated in the formation of porcine MP and lard diacylglycerol composite gels [41]. Moreover, with the increase in the amount of AMG, the hydrogen bonding content of frankfurters initially decreased and then increased (*p* < 0.05). Jiang, Ma, Wang, Wang, and Zeng indicated that the incorporation of κ-carrageenan induced a lower content of hydrogen bonds compared to the control group [42]. When the amount of AMG exceeded 0.2%, the hydrogen bond content began to increase gradually (*p* < 0.05), which was related to the crosslinking between the hydroxyl group in AMG and the polar amino acids in the side chain of MPs [43]. Yang, Wang, Li, and Chen found that a higher basil seed gum (BSG) content could increase the hydrogen bond content of a polysaccharide-protein system, which is responsible for the major hydroxyl groups of BSG that participate in the formation of hydrogen bonds [44].

The protein solubility in the 0.5% SDS solution of frankfurters supplemented with AMG was higher than that of the control group, indicating that hydrophobic interactions increased in the system (*p* < 0.05). This might be due to the transition from the compact α-helical structure of meat proteins to unfolded structures after the addition of AMG, which was beneficial for exposing protein hydrophobic groups [45]. Our result agreed with that of Zhang, Li, Wang, Xue, and Xue (2016), who reported that the hydrophobic interactions of surimi gels were remarkably increased due to the addition of KGM [46].

The disulphide bonds of frankfurters can be elucidated by dissolving them in 0.25% β-ME solution. When the AMG concentration ranged from 0.1% to 0.3%, the disulphide bond content of the frankfurters significantly decreased (*p* < 0.05). The reason for this is that the sulfhydryl groups in AMG increase steric hindrance, which hinders the sulfhydryl groups from converting to disulphide bonds [44]. Furthermore, the disulphide bond content began to increase gradually when the AMG concentration was higher than 0.3% (*p* < 0.05), which might be due to the covalent cross-linking between the higher AMG content and MPs [38]. Zhu et al. reported that the content of disulphide bonds in lamb myofibrillar protein gel significantly increased due to the addition of psyllium husk polysaccharide [47]. In addition, the results suggested that the samples treated with the 0.25% β-ME solution had higher solubility, implying that disulphide bonds were one of the important molecular forces for meat protein gelation.

The variations in the ionic bonds in the frankfurters can be determined by treating them with a 0.5 mol∙L^−1^ NaSCN solution. According to Table 6, the number of ionic bonds in frankfurters with AMG added increased significantly compared to the control group (*p* < 0.05). AMG is an acidic polysaccharide that has numerous negative charges that can combine with the opposite charges of the meat proteins, resulting in stronger electrostatic interactions in a complex gel matrix. Similar results were reported in our previous study, where Cao et al. found that κ-carrageenan clearly increased the content of ionic bonds in frankfurters [9]. However, higher concentrations of AMG (more than 0.3%) led to decreased ionic bonds in the frankfurters, which might be due to weakened electrostatic forces. The changing trend of ionic bonds was consistent with Zhou et al., who reported that the ionic bonds in myofibrillar protein gelation increased initially and then decreased due to the addition of psyllium husk gum [48]. Buda et al. also reported a similar trend in silver carp surimi gel in the presence of apple pectin [49].

Therefore, our findings suggested that hydrogen bonds were the dominant molecular forces in frankfurters with or without AMG. In addition, the molecular force results also demonstrated that the improvement of the quality profiles of the frankfurters with AMG was not only due to its “filling effect” but also due to the interaction between polysaccharide molecules and meat proteins.

#### 3.2.3. Microstructure

To further observe the gel network conformations of the meat proteins, the microstructure of the frankfurters was examined by SEM. As shown in Figure 3A, a loose and porous gel network structure with increasingly larger holes throughout (indicated by the white arrows in Figure 3A) was observed in the control group. This rough microstructure could not tightly trap water and fat during the heating process and could be easily broken by external forces, which explains the lower hardness, springiness, and *G′* value, as well as higher cooking loss, as discussed above. Zhuang et al. reported that there were more water channels throughout the MP gel net structure which might produce a negative impact on gel interaction [50]. Moreover, based on the previous texture and dynamic rheological behaviour results, the frankfurters were improved most effectively when the AMG addition concentration reached 0.3%. Therefore, the microstructure of the AMG-0.3% group was further observed and analysed by SEM. As shown in Figure 3B, frankfurters with AMG added showed a more continuous, uniform, and compact three-dimensional gel network structure compared to the control group, which was mainly attributed to more hydrophobic interactions and ionic bonds between the meat proteins and AMG. Zhao et al. reported that regenerated cellulose (RC) could induce the formation of a more uniform and denser MP gelation network structure, which might be due to the higher viscosity of the RC [51]. Zhuang, Wang, Jiang, Chen, and Zhou also demonstrated that a more compact MP network structure was obtained with the incorporation of KGM [52]. However, compared to the AMG-0.3% group, the MP gel networks loosened and formed a filamentous shape with an increased AMG addition of 0.5% (as shown in Figure 3C). The reason for this was that the higher content of polysaccharide restricted MP aggregation, causing a weaker gel network, which was in agreement with the texture analysis results. Zhuang et al. pointed out that excess KGM induced the formation of a weaker hydrogel by hydrogen bonds and was interpenetrated with meat protein before the heating process [53].

### 3.3. Digestive Properties of Proteins

#### 3.3.1. In Vitro Protein Digestibility

The in vitro digestibility of frankfurters with different formulations is shown in Figure 4A. When minced meat was digested through the in vitro stomach and intestines, the meat protein digestibility of the frankfurters in each group increased continuously with increasing time, implying that the meat protein was gradually degraded into small molecular peptides or amino acids under the action of pepsin and trypsin. According to Figure 4A, the protein digestibility of the frankfurters supplemented with AMG was clearly higher than that of the control group in both the stomach and intestine, and the digestibility increased with increasing AMG concentration (*p* < 0.05). This phenomenon might be because AMG easily absorbed water and swelled, which increased the effective contact areas between the meat proteins and enzyme, thus improving the digestibility of the meat proteins in the frankfurters. In addition, some functional polysaccharides have been shown to possess the ability to improve the activity of digestive enzymes when added to food in the gastrointestinal tract [12]. Saengsuk et al. reported that the incorporation of 0.5% low acyl gellan increased the protein digestibility of restructured pork steak [10]. Markussen et al. reported that milk protein concentrate was more evenly dispersed due to the addition of guar gum, with the formation of smaller protein aggregates, which increased the contact surface area with the enzyme, thus facilitating protein digestion [54]. Moreover, the conformational properties of meat proteins could also affect their digestibility. Jiang et al. found that the rigid structure of proteins was reduced (α-helix), while the disordered structure increased (random coil) due to the unfolding of meat proteins, which induced the protein structure to be more flexible and easier to combine with digestive enzymes, thus improving digestibility [18]. This result was consistent with previous changes in the secondary structure.

#### 3.3.2. Digestion Kinetics

In general, hydrocolloids produce a significant effect on the protein digestion kinetics of frankfurters. As shown in Figure 4B, the meat protein digestibility of frankfurters increased with digestion time due to the action of digestive enzymes. Moreover, the in vitro protein digestibility of frankfurters with AMG added was higher than that of the control group and increased with increasing AMG concentration. Furthermore, the in vitro digestibility of the control group appeared to plateau at the end of the digestion curve, which is consistent with the relatively high-rate constant (as shown in Table 7), indicating the rapid reach of the final plateau. In contrast, the samples with AMG added required more digestion time to settle down, which corresponded to a lower rate constant [55]. All the treated groups eventually showed a higher degree of protein digestion after 4 h of digestion than the control group (from 69.87% to 87.31%). The frankfurters with AMG added formed small and uniform protein particles which obtained a larger protease cleavage surface. In addition, AMG easily absorbed water and swelled, which increased the effective contact areas between the meat proteins and enzymes, thus improving the digestibility of the meat proteins in frankfurters.

#### 3.3.3. Particle Size

Particle size can represent the degree of protein decomposition, which is a crucial indicator to evaluate protein digestion [56]. As shown in Figure 5, the particle size of meat proteins before digestion decreased significantly with increasing concentrations of AMG (*p* < 0.05). This might be because the meat proteins were sufficiently unfolded due to AMG incorporation, which allowed the meat protein to be evenly dispersed in the extract solution, and thus the particle size to be reduced.

After gastric digestion, the particle size of the meat proteins in the frankfurters significantly decreased compared to the initial samples before digestion, which might be due to the decomposition of meat proteins by pepsin. Moreover, the particle size of the digested samples with AMG added was significantly decreased compared to the control group (*p* < 0.05) and gradually decreased with increasing AMG amount. This result implied that the addition of AMG promoted protein digestion in frankfurters during the gastric digestion stage. Similarly, Guevara-Zambrano et al. reported that the samples with higher digestibility had smaller particle sizes [57].

The particle size of the intestinal-digested samples was further decreased compared to the gastric-digested products, implying that the addition of trypsin could facilitate the further breakdown of proteins into small peptides [19]. Similar to the particle size results of the gastric digested samples, the particle size of the intestinal digested samples incorporated with AMG was significantly smaller than the control group (*p* < 0.05), and it gradually decreased after the addition of AMG.

#### 3.3.4. Particle Size Distribution and Super-Resolution Microscopy

Super-resolution microscopy images and the particle size distribution of the samples containing AMG at various concentrations were examined to monitor protein aggregation behaviour before and after digestion, as shown in Figure 5. As expected, before digestion, several aggregated protein particles with a broad particle distribution were observed in the microscopy images. With increasing AMG concentration, the protein particles tended to be more dispersed in the microscopic images with a narrower particle size distribution, which corresponded with the previous particle size results. After gastric digestion, the peaks of the particle size distribution gradually shifted to the left, implying a smaller particle size, which was in agreement with the particle size results. Concurrently, the particle size distribution ranges narrowed significantly, indicating that the protein particles were more evenly distributed, which could be observed in the microscopic images. This phenomenon suggested that pepsin decomposed the meat proteins into smaller peptides. Guevara-Zambrano et al. reported that the distribution of meat proteins was more uniform in the microstructure when protein digestibility was higher [57]. With increasing AMG concentration, the fluorescent red spots (representing meat proteins) gradually became finer and looser, suggesting a higher degree of protein degradation due to AMG addition. After intestinal digestion, the red spots became smaller and homogeneous, indicating that trypsin continued acting to convert the proteins into smaller peptides. Similar to the gastric digestion results, the meat protein samples containing AMG showed smaller aggregates with a more uniform dispersion compared to the control group, and the distribution peak gradually shifted to the left with a narrower range. In conclusion, in correspondence with the particle size results, these observations confirmed that the incorporation of AMG changed the conformational characteristics of the meat proteins, which made the proteins combine more easily with digestive enzymes, thus improving the protein digestibility of frankfurters.

### 3.4. Schematic Mechanism

The schematic mechanism for the improvement of frankfurter quality profiles induced by AMG is presented in Figure 6. The meat proteins formed a protein gel network with more water channels throughout it after heat treatment, which caused the loss of water and fat, thus leading to a higher cooking loss of the frankfurters. For the AMG-0.3% group, AMG could be used as a splendid “filler” to fill the water channels in the meat protein gel matrix, thereby trapping more water and fat, as well as reducing the cooking loss of frankfurters. Concurrently, the meat protein gel network structure veered towards greater uniformity and density due to the filling effect of AMG, which could improve the textural and rheological characteristics of the frankfurters. However, when the AMG addition level was increased to 0.5%, although the cooking loss of the frankfurters was further reduced due to the AMG filling more water channels, the overfilling resulted in the loss of the original springiness and chewiness of the frankfurters, leading to a decrease in the textural parameters. Moreover, the molecular force results confirmed that the improvement of the frankfurter quality profiles by AMG was not only due to the physical “filling effect”, but also contributed by the interaction between polysaccharide molecules and meat proteins. Furthermore, AMG easily absorbed water and swelled, which increased the effective contact area between meat proteins and digestive enzymes, thus improving the digestibility of the meat proteins in the frankfurters.

## 4. Conclusions

*Abelmoschus manihot* gum (AMG), as a new type of food gum, significantly enhanced the gel and quality profiles of frankfurters, as well as the dynamic rheological properties of uncooked meat batters, with the optimum effects being achieved when the concentration was 0.3%. This improvement due to the addition of AMG in frankfurters was not only due to its “filling effect”, but also to the interaction between polysaccharide molecules and meat proteins, which were demonstrated by the results of conformational characteristics. Furthermore, this study found that hydrogen bonds and disulphide bonds were the dominant molecular forces in frankfurters, even in the presence of AMG. Moreover, the incorporation of AMG significantly increased the in vitro digestibility of meat proteins in the frankfurters with increasing levels of addition, which was attributed to the physical and chemical effects between the meat proteins and AMG. Therefore, our present work exhibited substantial information on the effects of frankfurter quality profiles and in vitro digestibility with AMG as well as its mechanism, which is critical for the practical application of AMG in the processing of emulsified meat products. Our future work will focus on evaluating the gelling formation mechanism of myofibrillar proteins-AMG sols during heating treatment under the conditions applied in our present work.

## Figures and Tables

**Figure 1 foods-12-01507-f001:**
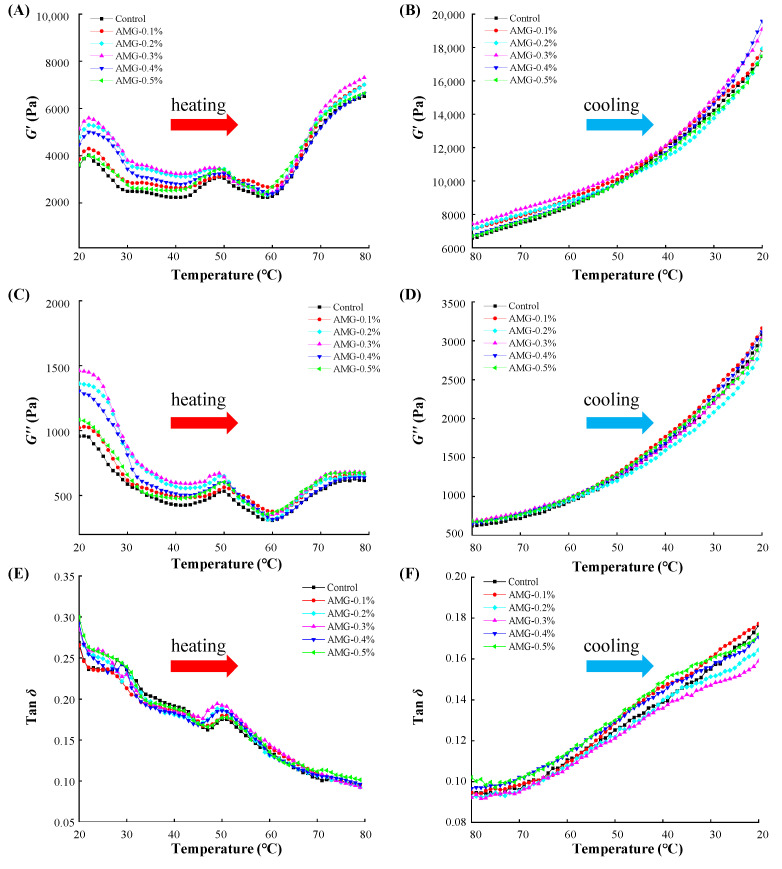
The rheological behaviour (storage modulus (*G′*); loss modulus (*G′′*); and loss tangent (tan *δ*)) of meat batters with or without AMG at different concentrations. (**A**,**C**,**E**) represent the heating treatment (from 20 °C to 80 °C). (**B**,**D**,**F**) represent the cooling treatment (from 80 °C to 20 °C). Control: no added AMG; AMG-0.1%: 0.1% added AMG; AMG-0.2%: 0.2% added AMG; AMG-0.3%: 0.3% added AMG; AMG-0.4%: 0.4% added AMG; and AMG-0.5%: 0.5% added AMG.

**Figure 2 foods-12-01507-f002:**
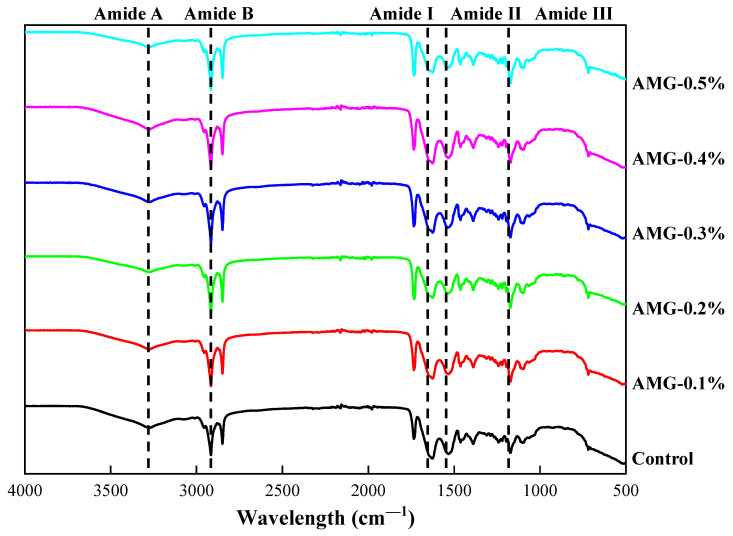
Infrared spectrogram of frankfurters with or without AMG at different concentrations. Control: no added AMG; AMG-0.1%: 0.1% added AMG; AMG-0.2%: 0.2% added AMG; AMG-0.3%: 0.3% added AMG; AMG-0.4%: 0.4% added AMG; and AMG-0.5%: 0.5% added AMG.

**Figure 3 foods-12-01507-f003:**
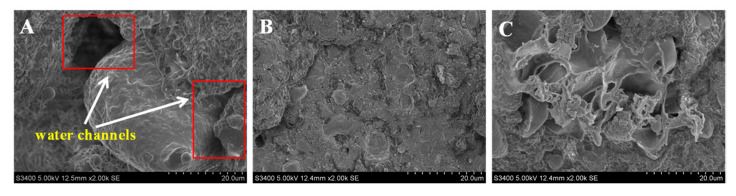
Scanning electron microscopy image of frankfurters with or without AMG at different concentrations. The magnification was 2000×. (**A**): Control group; (**B**): AMG-0.3% group; (**C**): AMG-0.5% group. Control: no added AMG; AMG-0.3%: 0.3% added AMG; and AMG-0.5%: 0.5% added AMG.

**Figure 4 foods-12-01507-f004:**
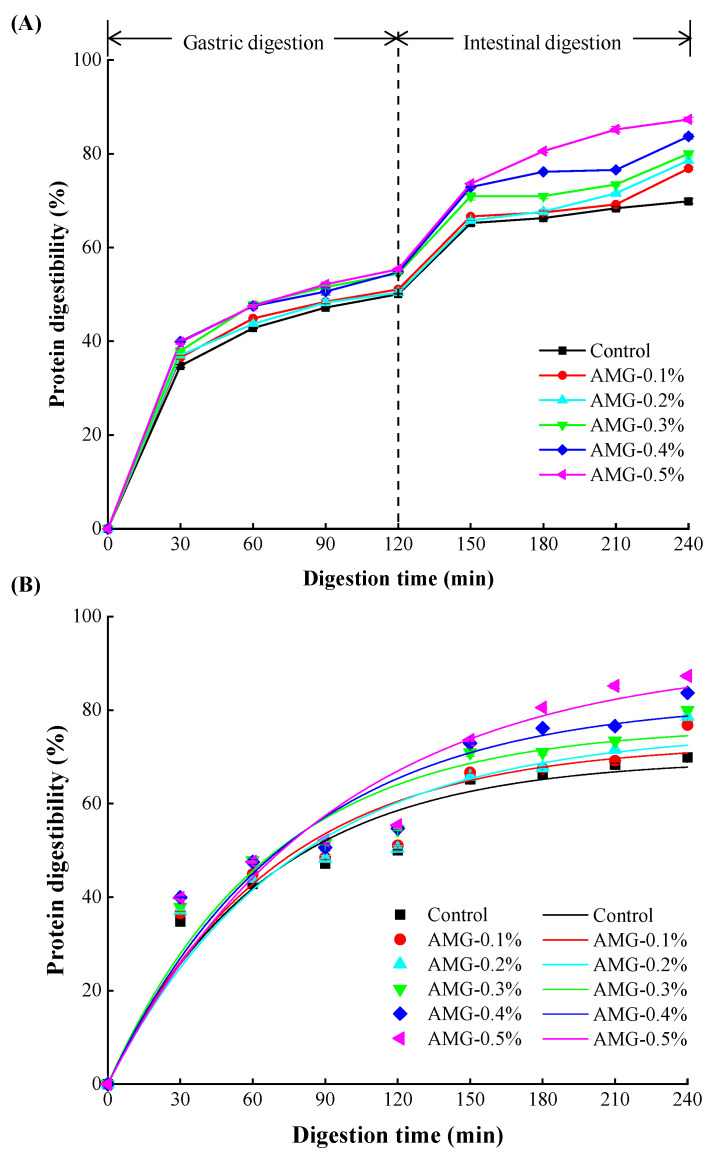
In vitro digestibility (**A**) and digestion kinetics (**B**) of meat proteins in frankfurters with or without AMG at different concentrations. Control: no added AMG; AMG-0.1%: 0.1% added AMG; AMG-0.2%: 0.2% added AMG; AMG-0.3%: 0.3% added AMG; AMG-0.4%: 0.4% added AMG; and AMG-0.5%: 0.5% added AMG.

**Figure 5 foods-12-01507-f005:**
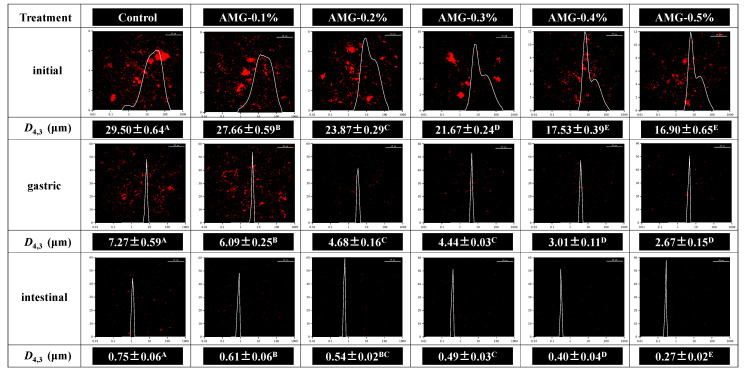
Microstructure and particle size distribution of the digestion samples of frankfurters with or without AMG at different concentrations. A–E in each column represent statistically significant differences (*p* < 0.05). Control: no added AMG; AMG-0.1%: 0.1% added AMG; AMG-0.2%: 0.2% added AMG; AMG-0.3%: 0.3% added AMG; AMG-0.4%: 0.4% added AMG; and AMG-0.5%: 0.5% added AMG.

**Figure 6 foods-12-01507-f006:**
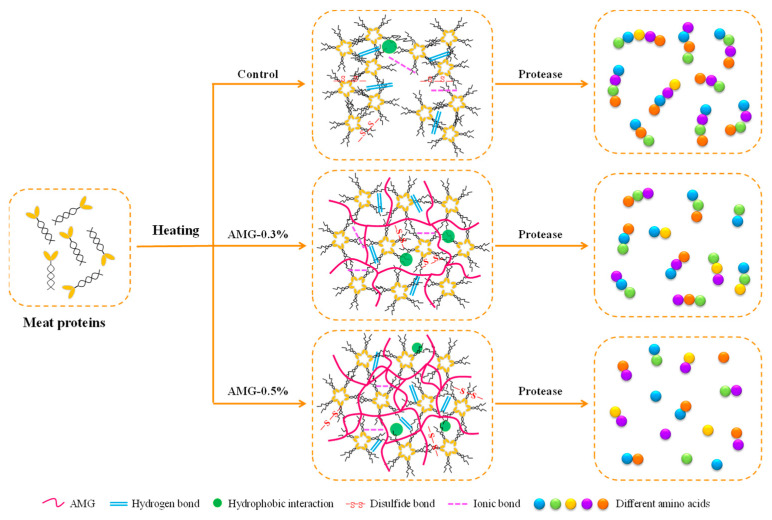
Mechanism diagram of the promoted quality profiles of frankfurters with or without AMG at different concentrations. Control: no added AMG; AMG-0.1%: 0.1% added AMG; AMG-0.2%: 0.2% added AMG; AMG-0.3%: 0.3% added AMG; AMG-0.4%: 0.4% added AMG; and AMG-0.5%: 0.5% added AMG.

**Table 1 foods-12-01507-t001:** Formulations of frankfurters with or without AMG at different concentrations.

	Control	AMG-0.1%	AMG-0.2%	AMG-0.3%	AMG-0.4%	AMG-0.5%
Lean meat (g)	500	500	500	500	500	500
Pork back fat (g)	250	250	250	250	250	250
Ice (g)	250	250	250	250	250	250
Total (g)	1000	1000	1000	1000	1000	1000
Sodium chloride (g)	15	15	15	15	15	15
Sodium nitrite (g)	0.05	0.05	0.05	0.05	0.05	0.05
Composite phosphates (g)	4	4	4	4	4	4
AMG (g)	0	1	2	3	4	5
Coriander seed powder (g)	0.5	0.5	0.5	0.5	0.5	0.5
Macis powder (g)	2.5	2.5	2.5	2.5	2.5	2.5
Red bell peppers powder (g)	2.5	2.5	2.5	2.5	2.5	2.5
Ginger powder (g)	3	3	3	3	3	3
White pepper powder (g)	3	3	3	3	3	3
Monosodium glutamate (g)	0.5	0.5	0.5	0.5	0.5	0.5
Sodium erythorbate (g)	1.52	1.52	1.52	1.52	1.52	1.52

Control: no added AMG; AMG-0.1%: 0.1% added AMG; AMG-0.2%: 0.2% added AMG; AMG-0.3%: 0.3% added AMG; AMG-0.4%: 0.4% added AMG; and AMG-0.5%: 0.5% added AMG.

**Table 2 foods-12-01507-t002:** The specific formulas of the stock solutions of simulated digestion fluids.

Constituent	Stock Conc. (mol L^−1^)	SGF (pH 3.0)		SIF (pH 7.0)	
		Vol. of Stock (mL)	Conc. in SGF (mmol L^−1^)	Vol. of Stock (mL)	Conc. in SIF (mmol L^−1^)
KCl	0.5	6.9	6.9	6.8	6.8
KH_2_PO_4_	0.5	0.9	0.9	0.8	0.8
NaHCO_3_	1.0	12.5	25	42.5	85
NaCl	2.0	11.8	47.2	9.6	38.4
MgCl_2_(H_2_O)_6_	0.15	0.4	0.1	1.1	0.33
(NH_4_)CO_3_	0.5	0.5	0.5	-	-
NaOH	1.0	-	-	-	-
HCl	6.0	1.3	15.6	0.7	8.4
CaCl_2_(H_2_O)_2_	0.3	-	0.15	-	0.6

**Table 3 foods-12-01507-t003:** Cooking loss (%) and emulsion stability (%) of frankfurters with or without AMG at different concentrations.

	Cooking Loss (%)	Emulsion Stability		
		Total Released Liquid (%)	Released Water (%)	Released Fat (%)
Control	2.59 ± 0.06 ^A^	0.899 ± 0.209 ^A^	0.818 ± 0.208 ^A^	0.081 ± 0.003 ^A^
AMG-0.1%	2.23 ± 0.09 ^B^	0.374 ± 0.032 ^B^	0.325 ± 0.033 ^B^	0.050 ± 0.002 ^B^
AMG-0.2%	2.12 ± 0.15 ^B^	0.286 ± 0.007 ^BC^	0.256 ± 0.011 ^BC^	0.030 ± 0.004 ^C^
AMG-0.3%	1.30 ± 0.16 ^C^	0.086 ± 0.001 ^CD^	0.068 ± 0.002 ^CD^	0.018 ± 0.001 ^D^
AMG-0.4%	1.19 ± 0.09 ^C^	0.011 ± 0.001 ^D^	0.006 ± 0.001 ^D^	0.006 ± 0.002 ^E^
AMG-0.5%	0.75 ± 0.01 ^D^	0.005 ± 0.001 ^D^	0.002 ± 0.002 ^D^	0.003 ± 0.002 ^E^

Results were expressed as the means ± SD based on triplicate measurements. A–E in each column represent statistically significant differences (*p* < 0.05). Control: no added AMG; AMG-0.1%: 0.1% added AMG; AMG-0.2%: 0.2% added AMG; AMG-0.3%: 0.3% added AMG; AMG-0.4%: 0.4% added AMG; and AMG-0.5%: 0.5% added AMG.

**Table 4 foods-12-01507-t004:** Textural properties of frankfurters with or without AMG at different concentrations.

	Hardness (g)	Resilience (g)	Springiness (%)	Fracturability (g)	Chewiness (g·s)	Adhesiveness (g·s)
Control	91.93 ± 2.71 ^D^	62.48 ± 0.91 ^BC^	60.49 ± 0.63 ^D^	301.56 ± 8.21 ^E^	1032.09 ± 3.56 ^C^	58.17 ± 0.32 ^B^
AMG-0.1%	98.71 ± 1.31 ^C^	62.08 ± 1.04 ^C^	61.74 ± 0.18 ^C^	332.56 ± 4.12 ^B^	1072.38 ± 13.85 ^B^	65.10 ± 0.27 ^A^
AMG-0.2%	102.86 ± 1.17 ^B^	64.24 ± 0.82 ^B^	62.43 ± 0.33 ^B^	343.67 ± 2.15 ^A^	1080.43 ± 2.81 ^B^	67.85 ± 0.02 ^A^
AMG-0.3%	106.42 ± 0.11 ^A^	66.36 ± 0.93 ^A^	63.67 ± 0.26 ^A^	346.47 ± 3.66 ^A^	1156.17 ± 12.42 ^A^	66.37 ± 2.09 ^A^
AMG-0.4%	102.19 ± 0.75 ^B^	64.37 ± 0.96 ^B^	62.01 ± 0.30 ^BC^	321.67 ± 1.56 ^C^	1069.76 ± 5.50 ^B^	57.51 ± 1.78 ^B^
AMG-0.5%	98.85 ± 0.32 ^C^	57.55 ± 1.54 ^D^	60.13 ± 0.43 ^D^	312.54 ± 2.28 ^D^	1006.14 ± 1.12 ^D^	54.52 ± 2.23 ^C^

Results were expressed as the means ± SD based on triplicate measurements. A–E in each column represent statistically significant differences (*p* < 0.05). Control: no added AMG; AMG-0.1%: 0.1% added AMG; AMG-0.2%: 0.2% added AMG; AMG-0.3%: 0.3% added AMG; AMG-0.4%: 0.4% added AMG; and AMG-0.5%: 0.5% added AMG.

**Table 5 foods-12-01507-t005:** The percentages (%) of the protein secondary structure of frankfurters with or without AMG at different concentrations.

	α-Helix	β-Sheet	β-Turn	Random Coil
Control	43.01 ± 0.36 ^A^	23.06 ± 0.09 ^D^	23.10 ± 0.14 ^D^	10.84 ± 0.13 ^E^
AMG-0.1%	41.60 ± 0.70 ^B^	23.24 ± 0.27 ^D^	24.16 ± 0.25 ^C^	10.99 ± 0.18 ^DE^
AMG-0.2%	40.28 ± 0.38 ^C^	23.85 ± 0.15 ^C^	24.49 ± 0.39 ^C^	11.38 ± 0.14 ^CD^
AMG-0.3%	39.18 ± 0.32 ^C^	24.29 ± 0.38 ^BC^	24.75 ± 0.23 ^BC^	11.80 ± 0.20 ^BC^
AMG-0.4%	37.89 ± 0.61 ^D^	24.65 ± 0.09 ^AB^	25.32 ± 0.34 ^AB^	12.17 ± 0.21 ^B^
AMG-0.5%	36.58 ± 0.64 ^E^	24.98 ± 0.10 ^A^	25.70 ± 0.36 ^A^	12.78 ± 0.21 ^A^

Results were expressed as the means ± SD based on triplicate measurements. A–E in each column represent statistically significant differences (*p* < 0.05). Control: no added AMG; AMG-0.1%: 0.1% added AMG; AMG-0.2%: 0.2% added AMG; AMG-0.3%: 0.3% added AMG; AMG-0.4%: 0.4% added AMG; and AMG-0.5%: 0.5% added AMG.

**Table 6 foods-12-01507-t006:** Solubility (%) of frankfurters with or without AMG at different concentrations in different force-disruption solutions.

	Urea	SDS	β-ME	NaSCN
Control	41.73 ± 0.60 ^C^	6.38 ± 0.09 ^C^	20.49 ± 0.24 ^A^	10.51 ± 0.28 ^C^
AMG-0.1%	39.36 ± 0.41 ^D^	6.45 ± 0.11 ^C^	18.81 ± 0.37 ^B^	12.44 ± 0.04 ^B^
AMG-0.2%	37.39 ± 0.30 ^E^	6.89 ± 0.07 ^B^	14.40 ± 0.38 ^C^	17.25 ± 0.20 ^A^
AMG-0.3%	41.86 ± 0.20 ^C^	7.08 ± 0.07 ^B^	12.95 ± 0.74 ^D^	17.67 ± 0.41 ^A^
AMG-0.4%	45.11 ± 0.29 ^B^	7.17 ± 0.07 ^AB^	15.23 ± 0.49 ^C^	11.04 ± 0.20 ^C^
AMG-0.5%	45.87 ± 0.44 ^A^	7.44 ± 0.21 ^A^	21.24 ± 0.98 ^A^	10.99 ± 0.30 ^C^

Results were expressed as the means ± SD based on triplicate measurements. A–E in each column represent statistically significant differences (*p* < 0.05). Control: no added AMG; AMG-0.1%: 0.1% added AMG; AMG-0.2%: 0.2% added AMG; AMG-0.3%: 0.3% added AMG; AMG-0.4%: 0.4% added AMG; and AMG-0.5%: 0.5% added AMG.

**Table 7 foods-12-01507-t007:** Kinetic parameters of the in vitro digestion of frankfurters with or without AMG at different concentrations.

	C_f_ (%)	k (min^−1^)	R^2^
Control	69.58 ± 4.22	0.015 ± 0.002	0.952
AMG-0.1%	72.97 ± 5.22	0.014 ± 0.003	0.938
AMG-0.2%	75.60 ± 6.33	0.013 ± 0.003	0.932
AMG-0.3%	76.57 ± 4.99	0.015 ± 0.003	0.946
AMG-0.4%	82.19 ± 6.91	0.013 ± 0.003	0.932
AMG-0.5%	91.35 ± 9.19	0.011 ± 0.002	0.928

C_f_ (%) is the final protein digestibility after 4 h digestion. k (min^−1^) is the digestion rate constant. Values are expressed as the estimated kinetic value ± SD of the estimated kinetic parameters from the model. Control: no added AMG; AMG-0.1%: 0.1% added AMG; AMG-0.2%: 0.2% added AMG; AMG-0.3%: 0.3% added AMG; AMG-0.4%: 0.4% added AMG; and AMG-0.5%: 0.5% added AMG.

## Data Availability

The data presented in this study are available in the article.

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
