# Peer review of "In-Depth Insight into the Mechanism of Incorporation of *Abelmoschus manihot* Gum on the Enhancement of Gel Properties and In Vitro Digestibility of Frankfurters"

_foods, 2023, doi:10.3390/foods12071507_

Round 1

Reviewer 1 Report

The article entitled "In-depth insight into the mechanism of incorporation of Abelmoschus manihot gum on the enhancement of gel properties and in vitro digestibility of frankfurters" is an interesting article which provides valuable information about the effects of the inclusion of different concentrations (0-0.5%) of Abelmoschus manihot gum (AMG) on frankfurters quality profiles and in vitro digestibility and its mechanism. This information is essential for future use in the food industry of AMG, with the importance that this entails. The introduction is well planned, with sufficient and suitable bibliography. At the same time, the exposition and discussion of the results, as well as the conclusions are shown in a very clear and adequate way. 

Lines 59: "Abelmoschus manihot" in italics.

Lines 69: "In vitro" in italics? Keep the same format throughout the entire manuscript.

Line 97-98: Although you cite the reference that describes the methodology used for the Frankfurter procedure, it would be appropriate to make a brief description in the text, since colleagues may not have access to the information at that time.

Line 101: Remove "g" after each number, as it is repetitive and add it in parentheses after the "Ingredient/ additive" [e.g., "Lean meat (g) "].

Line 105-107: In the same way as for the manufacturing process of the Frankfurters, it briefly explains the methodology followed to determine the cooking loss and the emulsion stability, despite also naming references. Act accordingly for the rest of the section.

Author Response

Response to Reviewer #1

The article entitled "In-depth insight into the mechanism of incorporation of Abelmoschus manihot gum on the enhancement of gel properties and in vitro digestibility of frankfurters" is an interesting article which provides valuable information about the effects of the inclusion of different concentrations (0-0.5%) of Abelmoschus manihot gum (AMG) on frankfurters quality profiles and in vitro digestibility and its mechanism. This information is essential for future use in the food industry of AMG, with the importance that this entails. The introduction is well planned, with sufficient and suitable bibliography. At the same time, the exposition and discussion of the results, as well as the conclusions are shown in a very clear and adequate way.

Q1Lines 59: "Abelmoschus manihot" in italics.

A1: This is a good question. First of all, we were apologized for the unclear or unprecise statement in our initial manuscript. According to the reviewer’s opinion, we have revised “Abelmoschus manihot” in italics.

We are indebted to the reviewer for this constructive suggestion to improve the quality of the manuscript. Thanks.

Q2: Lines 69: "In vitro" in italics? Keep the same format throughout the entire manuscript.

A2: This is a good question. First of all, First of all, we were apologized for the unclear or unprecise statement in our initial manuscript. According to the reviewer’s opinion, we have revised the “In vitro” in italics throughout the entire manuscript in the revised manuscript.

We are indebted to the reviewer for this constructive suggestion to improve the quality of the manuscript. Thanks.

Q3: Line 97-98: Although you cite the reference that describes the methodology used for the Frankfurter procedure, it would be appropriate to make a brief description in the text, since colleagues may not have access to the information at that time.

A3: This is a good suggestion. Firstly, we are apologized for the incomplete statement about the preparation of frankfurter in our initial manuscript. According to the reviewer’s opinion, we have revised the information about the detailed testing procedure as follows:

2.2 Preparation of frankfurters

The fundamental formulations of the different frankfurters are shown in Table 1. Different levels of AMG incorporation (0.0, 0.1, 0.2, 1.3, 0.4, and 0.5%, w/w, based on the total weight of the samples) were added into meat batters and marked as Control, AMG-0.1%, AMG-0.2%, AMG-0.3%, AMG-0.4%, and AMG-0.5% respectively. The detailed processing procedure of the frankfurters was performed according to the method of Yuan et al. [13]. Briefly, all visible connective tissue was removed from the raw meat before the preparation of meat batters. Then, the pork shoulder meat and pork back fat were respectively ground through 8 mm and 3 mm plate via a mincer (BJRJ-82, Xinghe Mechanical Co., Ltd., Nantong, Jiangsu, China). The pork lean meat was added to the bowl chopper (BZBJ-20, Xinghe Mechanical Co., Ltd., Nantong, Jiangsu, China) and chopped for 2 min with salt, sodium nitrite, composite phosphates and half of the ice. Then, the pork backfat, different concentrations of AMG, spices and the remaining ice were added to the meat batters and chopped for 3 min. After that, sodium erythorbate was added to the meat batters and chopped for 1 min. The entire temperature of meat batters was below 12 °C in all instances. The meat batters were stuffed into collagen casings with 20 mm diameter. The frankfurters were transferred to an automatic smoking chamber (BYXX-50L, Xinghe Mechanical Co., Ltd., Nantong, Jiangsu, China) for drying and smoking. All the different prepared frankfurter samples were vacuum packed and stored in the refrigerator at 4 °C until the instrumental measurements were finished.”

We are indebted to the reviewer for this constructive suggestion to improve the quality of the manuscript. Thanks.

Q4: Line 101: Remove "g" after each number, as it is repetitive and add it in parentheses after the "Ingredient/ additive" [e.g., "Lean meat (g) "].

A4: This is a good suggestion. First of all, we were apologized for the unclear statement in our initial manuscript. According to the reviewer’s opinion, we have deleted the “g” after each number in our revised manuscript.

We are indebted to the reviewer for this constructive suggestion to improve the quality of the manuscript. Thanks.

Q5: Line 105-107: In the same way as for the manufacturing process of the Frankfurters, it briefly explains the methodology followed to determine the cooking loss and the emulsion stability, despite also naming references. Act accordingly for the rest of the section.

A5: This is a good suggestion. Firstly, we are apologized for the unclear statement about the measurement about cooking loss and the emulsion stability in our initial manuscript. According to the reviewer’s opinion, we have added the related information in our revised manuscript as follows:

“The cooking loss (%) of the different frankfurters was determined using the method of Cao et al. [9]. Briefly, the cooked frankfurters (length and diameter of 15 cm and 20 mm, respectively) were refrigerated overnight at 4 °C and then weighed. The cooking loss (%) was expressed of the weights of cooked frankfurters relative to the weights of un-cooked frankfurters, and the calculation formula was as follows:

Where W1 was the weight of un-cooked frankfurters (g), W2 was the weight of cooked frankfurters (g)”

“The emulsion stability of the different frankfurters was measured and expressed according to the same procedure described by Zhang et al. [14]. Briefly, approximately 35 g of meat batter was weighed in tubes and then centrifuged (3500 g, 5 min, at 4 °C). Subsequently, the tubes were heated at 80 °C in the water bath for 30 min and then left to stand upside down for 1 h to release the juice to the pre-weighed aluminum dish. Then, the aluminum dish was weight and placed in the 105 °C baking oven (DHG-9030A, Xinghe Mechanical Co., Ltd., Nantong, Jiangsu, China) and baked until the weight was constant. The expression of emulsion stability were as follows: Total released liquid (%) was expressed as the ratio of the weight of the released juice after inversion for 1 h to the total weight of the meat batter. Released water (%) was expressed as the ratio of weight lost after baking to the total weight of meat batter. Released fat (%) was expressed as the ratio of the weight left after baking to the total weight of the meat batter. ”

We are indebted to the reviewer for this constructive suggestion to improve the quality of the manuscript. Thanks.

Reviewer 2 Report

Topic of manuscript is interesting, valuable and relevant for the field, up of date and is within the thematic scope of the section of “Meat” in Molecules journal (and scope of special issue “Comprehensive Approaches to Formulation of Meat Products: From Technological Development to Assessment of Healthy Properties”).

 The manuscript reports quite interesting results obtained by systematic investigation of the samplers prepared in a highly professional manner. The content of the paper promises some commercial application of the Abelmoschus manihot gum as functional ingredient in frankfurters suggested in this paper. I have the following suggestions for the improvement of the paper.

Detailed comments:

1.     Part 2.1. please supply more information about the meat origin which is crucial ingredient of frankfurters (breed of pigs etc).

2.     Tables and Figures: Please consider to include explanations of all acronyms under the Tables in small Font (variants of samples), the Table should be selfredable.

3.     In part 2.3. Please include one sentence describing procedure of calculation of cooking loss and emulsion stability.

4.     In part 2.4. please supply analysed parameters of texture.

5.     Part 2.9. please insert more description in explanation of method.

6.     Part 3.1.1. Please supply more explanation about probably reason of reducing cooking loss experimental samples contained AMG.

7.     Regarding discussion results of texture you obtained that some level of AMG caused increase of values texture and some level imparted decrease of these values. Which values of texture and content of AMG would be more preferable for consumers ? please supply one/two sentence/s.

8.     Conclusion needs some reformulation/supplementation. Firstly, avoid to use acronyms which make it difficult fast understanding for future readers. Maybe insert explanation of AMG acronym. What is the future of your findings?

Minor comments

Lines - 43, 63, 69: please supply the year of publishing of cited paper.

Author Response

Response to Reviewer #2

Topic of manuscript is interesting, valuable and relevant for the field, up of date and is within the thematic scope of the section of “Meat” in Molecules journal (and scope of special issue “Comprehensive Approaches to Formulation of Meat Products: From Technological Development to Assessment of Healthy Properties”).

The manuscript reports quite interesting results obtained by systematic investigation of the samplers prepared in a highly professional manner. The content of the paper promises some commercial application of the Abelmoschus manihot gum as functional ingredient in frankfurters suggested in this paper. I have the following suggestions for the improvement of the paper.

Q1Part 2.1. please supply more information about the meat origin which is crucial ingredient of frankfurters (breed of pigs etc).

A1: This is a good question. Firstly, we are apologized for the unclear and inaccurately statement about the meat origin in our initial manuscript. According to the reviewer’s opinion, we have added the essential information as follows:

“2.1. Materials and chemicals

Post-rigor lean pork shoulder meat (moisture 73.90%, protein 21.36%, and fat 3.17%, based on total weight) and pork back fat (moisture 7.46% and fat 86.95%, based on total weight) from northeast min pig were purchased from Beidahuang Meat Co., Ltd. (Suihua, Heilongjiang, China) and stored at 4 °C while delivered to the lab and used within the same day. Food-grade AMG was kindly provided by Huaxing Food Additives Co., Ltd. (Jinan, Shandong, China). Pepsin (P7012, ≥ 400 units/mg) and trypsin (S31655, ≥ 2500 units/mg) were purchased from Sigma-Aldrich (St. Louis, MO, USA). All spices were purchased from Vemis Spices Co., Ltd. (Taizhou, Jiangsu, China).”

We are indebted to the reviewer for this constructive suggestion to improve the quality of the manuscript. Thanks.

Q2: Tables and Figures: Please consider to include explanations of all acronyms under the Tables in small Font (variants of samples), the Table should be selfredable.

A2: This is a good suggestion. Firstly, we are apologized for the unclear statements in our initial manuscript. According to the reviewer’s opinion, we have added the explanations of all acronyms under the Tables and Figures in our revised manuscript.

We are indebted to the reviewer for this constructive suggestion to improve the quality of the manuscript. Thanks.

Q3: In part 2.3. Please include one sentence describing procedure of calculation of cooking loss and emulsion stability.

A3: This is a good suggestion. Firstly, we are apologized for the unclear statement about the measurement about cooking loss and the emulsion stability in our initial manuscript. According to the reviewer’s opinion, we have added the related information in our revised manuscript as follows:

“The cooking loss (%) of the different frankfurters was determined using the method of Cao et al. [9]. Briefly, the cooked frankfurters (length and diameter of 15 cm and 20 mm, respectively) were refrigerated overnight at 4 °C and then weighed. The cooking loss (%) was expressed of the weights of cooked frankfurters relative to the weights of un-cooked frankfurters, and the calculation formula was as follows:

Where W1 was the weight of un-cooked frankfurters (g), W2 was the weight of cooked frankfurters (g).”

“The emulsion stability of the different frankfurters was measured and expressed according to the same procedure described by Zhang et al. [14]. Briefly, approximately 35 g of meat batter was weighed in tubes and then centrifuged (3500 g, 5 min, at 4 °C). Subsequently, the tubes were heated at 80 °C in the water bath for 30 min and then left to stand upside down for 1 h to release the juice to the pre-weighed aluminum dish. Then, the aluminum dish was weight and placed in the 105 °C baking oven (DHG-9030A, Xinghe Mechanical Co., Ltd., Nantong, Jiangsu, China) and baked until the weight was constant. The expression of emulsion stability were as follows: Total released liquid (%) was expressed as the ratio of the weight of the released juice after inversion for 1 h to the total weight of the meat batter. Released water (%) was expressed as the ratio of weight lost after baking to the total weight of meat batter. Released fat (%) was expressed as the ratio of the weight left after baking to the total weight of the meat batter. ”

We are indebted to the reviewer for this constructive suggestion to improve the quality of the manuscript. Thanks.

Q4: In part 2.4. please supply analysed parameters of texture.

A4: This is a good suggestion. This is a good suggestion. Firstly, we are apologized for the unclear statement about the measurement parameters about texture in our initial manuscript. According to the reviewer’s opinion, we have added the related information in our revised manuscript as follows:

“2.4. Texture

According to the detailed testing procedures and parameters of Yuan et al. [15], two deformation tests were used to determine the texture of frankfurters with a TA-TX plusC texture analyser (Stable Micro Systems Co., Ltd., Godalming, UK). The detailed testing parameters were as follows: the pre-test speed, test speed, and post-test speed were 1.5 mm/s, 1.5 mm/s, and 10.0 mm/s, respectively. The trigger force was 15.0 g. Moreover, the testing procedure was divided into two consecutive cycles as follows: (1) the first cycle was not break into the surface of frankfurters at 15.0% strain, which mainly reflected the hardness (g), springiness (%) and resilience (g); (2) the second cycle was puncture into the interior of frankfurters at 75.0% strain, which mainly reflected the chewiness (g.s), fracturability (g) and adhesiveness (g.s). And the holding time between two cycles was 5 s.”

We are indebted to the reviewer for this constructive suggestion to improve the quality of the manuscript. Thanks.

Q5: Part 2.9. please insert more description in explanation of method.

A5: This is a good suggestion. Firstly, we are apologized for the unclear statement about the description for In vitro gastrointestinal digestion methods in our initial manuscript. According to the reviewer’s opinion, we have added the related information in our revised manuscript as follows:

“2.9. In vitro gastrointestinal digestion

The frankfurters were minced and mixed with phosphate-buffered saline (5 mL, 10 mmol/L Na2HPO4–NaH2PO4, pH 7.0). The mixed solution was treated with pepsin and trypsin to simulate the human gastrointestinal digestion process. Then the steps of gastric and intestinal digestion were according to the steps in 2.9.1 and 2.9.2, respectively. In addition, the specific formulations of simulated gastric fluid (SGF) and simulated intestinal fluid (SIF) were prepared according to method of Jiang et al. [18], and the specific formulas of SGF and SIF were according to the Table 2.”

We are indebted to the reviewer for this constructive suggestion to improve the quality of the manuscript. Thanks.

Q6: Part 3.1.1. Please supply more explanation about probably reason of reducing cooking loss experimental samples contained AMG.

A6: This is a good suggestion. Firstly, we are apologized for the unclear statement about the reason of reducing cooking loss in our initial manuscript. In fact, the reason of reducing cooking loss was as follows: firstly, AMG had the superior ability of absorbing water, which could absorb more water and fat in meat batters, thus reducing the cooking loss. In addition, the addition of AMG promoted the formation of fine and uniform gel network structure, thus trapping more water and fat. Therefore, according to the reviewer’s opinion, we have added some statements in our revised manuscript as follows:

“Cooking loss is one of the basic indicators to detect the quality attributes of meat products, which could reflect the water and oil holding capacity of the frankfurters in this study. As shown in Table 2, the cooking loss of frankfurters added with AMG decreased significantly compared to the control group (P < 0.05), indicating that more water and oil were retained in the meat protein matrix due to the addition of AMG. With increasing AMG concentrations from 0.1% to 0.5%, the cooking loss of the frankfurters decreased by 13.90, 18.15, 49.81, 54.05, and 71.04%. This finding was consistent with that of Barbut and Somboonpanyakul, who reported that the incorporation of crude malva nut gum significantly reduced the cooking loss of emulsified chicken meat batter [21]. The reason for this phenomenon was that AMG had superior water absorption, which could absorb more water and fat, thus reducing the cooking loss. In addition, the addition of AMG promoted the formation of fine and uniform gel network structure, thus trapping more water and fat. In addition, Han et al. pointed out that inulin could alter the interaction between water molecules and MPs and strengthen the network structure of the protein gel, thus enhancing the water-holding capacity of the MP gel [22].”

We are indebted to the reviewer for this constructive suggestion to improve the quality of the manuscript. Thanks.

Q7: Regarding discussion results of texture you obtained that some level of AMG caused increase of values texture and some level imparted decrease of these values. Which values of texture and content of AMG would be more preferable for consumers ? please supply one/two sentence/s.

A7: This is a good question. Firstly, we are apologized for the unclear statement about the appropriate content of AMG for consumers in our initial manuscript. In fact, the 0.3% of AMG was the most appropriate concentration to improve the texture of frankfurters. According to the reviewer’s opinion, we have added the statement as follows:

“Therefore, the texture results implied that a suitable AMG addition standard was a fea-sible strategy to improve the textural features of frankfurters, especially for the AMG concentration of 0.3%.”

We are indebted to the reviewer for this constructive suggestion to improve the quality of the manuscript. Thanks.

Q8: Conclusion needs some reformulation/supplementation. Firstly, avoid to use acronyms which make it difficult fast understanding for future readers. Maybe insert explanation of AMG acronym. What is the future of your findings?

A8: This is a good question. Firstly, we are apologized for the unclear statement about the conclusion in our initial manuscript. According to the reviewer’s opinion, we have revised the statement in the section of “conclusion” as follows:

Abelmoschus manihot gum (AMG), as a new type of food gum, significantly facilitated the gel and quality profiles of frankfurters as well as the dynamic rheological properties of uncooked meat batters, and the optimum effects were achieved when the concentration was 0.3%. This improvement of AMG in frankfurters was not only due to its “filling effect”, but also to the interaction between polysaccharide molecules and meat proteins, which have been demonstrated by the results of conformational characteristics. Furthermore, this study found that hydrogen bonds and disulphide bonds were the dominant molecular forces in frankfurters, even in the presence of AMG. Moreover, the incorporation of AMG significantly increased the in vitro digestibility of meat proteins in frankfurters with increasing level of addition, which was attributed to the physical and chemical effects between meat proteins and AMG. Therefore, our present work exhibited substantial information on the effects of frankfurter quality profiles and in vitro digestibility by AMG and its mechanism, which is critical for the actual application of AMG in the processing of emulsified meat products. Our future work will focus on evaluating the gelling formation mechanism of myofibrillar proteins-AMG sols during the heating treatment under the conditions applied in our present work.”

We are indebted to the reviewer for this constructive suggestion to improve the quality of the manuscript. Thanks.

Q9: Lines - 43, 63, 69: please supply the year of publishing of cited paper.

A9: This is a good suggestion. Firstly, we are apologized for the unclear and unprecise statement about the year of publishing of cited paper in our initial manuscript. According to the reviewer’s opinion, we have added the information about the year of of publishing of cited paper in our revised manuscript.

We are indebted to the reviewer for this constructive suggestion to improve the quality of the manuscript. Thanks.

Round 2

Reviewer 2 Report

The manuscript was sufficiently revised by the Authors. I also appreciate responses on all comments.